# DNA-Barcoding for Cultivar Identification and Intraspecific Diversity Analysis of Agricultural Crops

**DOI:** 10.3390/ijms26146808

**Published:** 2025-07-16

**Authors:** Lidiia S. Samarina, Natalia G. Koninskaya, Ruset M. Shkhalakhova, Taisiya A. Simonyan, Daria O. Kuzmina

**Affiliations:** Federal Research Centre the Subtropical Scientific Centre of the Russian Academy of Sciences, 354002 Sochi, Russia; natakoninskaya@mail.ru (N.G.K.); shhalahova1995@mail.ru (R.M.S.); taisiya-simony@yandex.ru (T.A.S.); kuzminad16@gmail.com (D.O.K.)

**Keywords:** cultivar identification, barcoding libraries, chloroplast DNA, mitochondrial DNA, intraspecific diversity, genetic passport, next generation sequencing, bioinformatic tools

## Abstract

DNA barcoding of intraspecific diversity of agricultural crops is important to develop the genetic passports of valuable genotypes and cultivars. The advantage of DNA-barcoding as compared to traditional genotyping of cultivars is that the procedure can be unified and applied for the broad range of accessions. This not only makes it cost efficient, but also allows to develop open access genetic databases to accumulate information of the world’s germplasm collections of different crops. In this regard, the aim of the review was to analyze the latest research in this field, including the selection of loci, universal primers, strategies of amplicons analysis, bioinformatic tools, and the development of databases. We reviewed the advantages and disadvantages of each strategy with the focus of cultivars identification. The data indicates that following chloroplast loci are the most prominent for the intraspecific diversity analysis: (trnE-UUC/trnT-GUU, rpl23/rpl2.l, psbA-trnH, trnL-trnF, trnK, rpoC1, ycf1-a, rpl32-trnL, trnH-psbA and matK). We suggest that the combination of three or four of these loci can be a sufficient DNA barcode for cultivar-level identification. This combination has to be selected for each crop. Advantages and disadvantages of different approaches of amplicons analysis are discussed. The bioinformatic tools and databases for the plant barcoding are reviewed. This review will be useful for selecting appropriate strategies for barcoding of intraspecific diversity of agricultural crops to develop genetic passports of valuable cultivars in germplasm collections worldwide.

## 1. Introduction

DNA barcoding, also known as genetic barcoding, is a molecular identification method that allows us to distinguish taxa using several standardized regions of DNA [1]. This method is used to characterize existing biodiversity to identify new species and genotypes [2]. The development of DNA barcodes for valuable cultivars of local germplasm collections is important for their tracking and analyzing genetic relationships. In addition, DNA barcoding is an important tool for studying the evolution, ecology, and conservation of plants, especially given that biodiversity is threatened by anthropogenic activities and unfavorable climatic factors [3,4,5].

The use of DNA barcoding for the identification and conservation of valuable varieties has attracted considerable attention in recent years, especially due to the strengthening of phytosanitary measures and biodiversity management. Most of the previously published reviews on the topic of DNA barcoding are devoted to plant identification at the species level [5,6,7,8,9,10,11]. To address the problem of cultivar identification, it is necessary to review recently published DNA barcoding studies that were efficient at the genotype level. In this regard, the aim of the current review was to analyze the latest research in this field, including the selection of loci, primers, strategies of amplicons analysis, bioinformatic tools, and databases development.

The advantage of DNA-barcoding as compared to traditional genotyping of cultivars is that the procedure can be unified and applied for the broad range of accessions. This not only makes it cost efficient, but also allows to develop open access genetic databases to accumulate information of the world’s germplasm collections of different crops. The cultivars’ barcoding is important for the development of genetic passports for molecular identification of cultivars of different agricultural crops. Genetic passports will help protect the copyrights of breeders, protect farmers from buying counterfeit products, and obtain reliable and complete information about a particular genotype. Due to the increasing number of cases of illegal commercialization of selected cultivars, the protection of plant breeders’ rights has become of paramount importance for breeding companies [12]. For example, single-nucleotide variants were identified among 15 lavender varieties and allowed for the assessment of the genetic identities of individual to protect Mediterranean breeding lines and confirm their origin [12]. 

The development of genetic passports of valuable genotypes depends on the selection of reliable strategies for DNA barcoding, including the correct choice of the target loci, suitable primers for particular crop, amplicon analysis strategy, bioinformatic tools, and the development of the databases of plant ex situ collections (Figure 1). Further we discuss each of these steps to help to select the efficient methodology which provides accurate, time- and labor-saving protocols for intraspecific diversity characterization of valuable crops. This review will be of interest for researchers who deal with germplasm collections, characterization, and breeding research.

## 2. Selection of an Appropriate Loci

The selection of a loci for the barcoding should follow the criteria of universality and high discriminatory power. Universality means that the locus is efficient for a wide range of species and genera, while discriminatory power is the ability to identify closely related genotypes. Methods based on mitochondrial DNA (mtDNA), chloroplast DNA (cpDNA), and nuclear DNA are used for barcoding land plants. Furthermore, we discuss the advantages, disadvantages, and some examples of them for cultivars identification.

### 2.1. Mitochondrial DNA for Cultivars Identification

Mitochondrial DNA has certain advantages, such as maternal inheritance and the potential for whole-genome resequencing. However, its application in plant cultivar barcoding is limited by low variability and structural conservatism, which reduce their discriminatory power at the cultivar level [13,14]. Consequently, the usefulness of mitochondrial DNA for plant cultivar identification remains controversial, with some studies pointing to limited resolution at the cultivar level. Moreover, a recent study reported that 35% of the ancestral plastid genomes were transferred to mitochondrial genomes over the past 10 million years. The authors reported that some plastid barcoding markers co-amplified the mitochondrial DNA and caused a mis-authentication of plants [15].

Despite this, few recent studies show the efficiency of mtDNA as an alternative or complementary loci for cultivar level identification. Particularly, the cox2 intron II region of mtDNA was efficiently used for the characterization of intraspecific diversity of *Panax ginseng* [16]. Additionally, whole mitochondrial genome analysis allowed for the identification of 156 unique mitochondrial SNPs in the collection of 171 cultivars of Tunisian date palm cultivars [14]. We have not found any other examples that used mtDNA for cultivar identification. Meanwhile, the analysis of complete mitochondrial genomes can probably be efficient to reveal polymorphisms at low taxonomic levels.

### 2.2. Chloroplast DNA for Cultivars Identification

The chloroplast genome, a closed circular DNA molecule present in multiple copies in plant cells [17], exhibits a high copy number, making it readily accessible for analysis, even from degraded samples [18]. The cpDNA contains a combination of conserved and variable regions useful for both phylogenetic studies and barcoding, allowing for the differentiation of closely related cultivars [14,19]. In 2009, a large group of plant DNA barcoding specialists proposed chloroplast genes *rbcL* and *matK* as the universal barcode. However, this universal barcode was often insufficient for intraspecific diversity analysis and had been met with great skepticism since its proposal [20]. Later, several other loci of cpDNA were proposed for plant DNA barcoding [21,22,23,24,25,26]. In 2011, the polymorphism of several plastid loci (rpl23&rpl2.1, 16S, 23S, 4.5S&5S, petB&D and rpl2, rpoC1 and trnK introns) was checked in 96 different plant species, and it was found that the trnK and rpoC1 introns are the most variable loci in closely related species, showing their high potential for barcoding [23]. Our recent results on four subtropical crops (actinidia, feijoa, mandarins, and tea) confirmed the efficiency of several chloroplast loci (23S,4.5S/5S, 16S, rpl23/rpl2.l, rpoC1 intron, trnE-UUC/trnT-GUU) for the cultivars’ barcoding. Among them, rpl23/rpl2.l and trnE-UUC/trnT-GUU showed the highest intraspecific polymorphisms for all crops, while rpl2 intron and 16S displayed the lowest polymorphism levels in the generated fragments [27].

The intergenic regions of the cpDNA are known for their greater polymorphism as compared to genes. According to Dong et al. [28], analysis of chloroplast genomes in 12 angiosperm genera revealed the top 5% most variable loci. The 23 loci in which the most variable were, in order from highest to lowest variability, the intergenic regions ycf1-a, trnK, rpl32-trnL, and trnH-psbA, followed by trnSUGA-trnGUCC, petA-psbJ, rps16-trnQ, ndhC-trnV, ycf1-b, ndhF, rpoB-trnC, psbE-petL, and rbcL-accD. Three loci, trnSUGA-trnGUCC, trnT-psbD, and trnW-psaJ, showed very high nucleotide diversity per site across three genera [28]. Another study reported that the trnH-psbA barcode region showed greater resolving power compared to *rbcL* and *matK* in several plant species [29]. However, some intergenic regions are not always efficient enough for cultivars identification. Particularly, the trnH-psbA and trnL-trnF intergenic spacers were able to distinguish and identify only four among 25 *Prunus* accessions [30]. Another research showed that trnH–psbA is not suitable for some taxa due to intraspecific inversions and rps19 insertions, which inflate intraspecific variation [31]. Particularly, trnH-psbA was efficient for the discrimination of *Physalis* species, but intraspecific polymorphism was low [32]. In addition, most of the intergenic regions were not efficient for *Triticum* barcoding. However, a combination of the intergenic regions trnfM-trnT with either trnD-psbM, petN-trnC, matK-rps16, or rbcL-psaI demonstrated a very high discrimination capacity [33]. These results confirm the necessity of crop-specific selection of loci for the cultivars’ barcoding. To summarize, many researchers have effectively used cpDNA in cultivar identification of various plant species, including *Vitis* [19], *Medicago sativa* [34], *Scutellaria baicalensis* [35], *Chrysanthemum* [36], *Phoenix* [13], *Orchidaceae* [37], and *Ficus* [38]. These examples demonstrate the broad applicability of cpDNA barcoding across various plant families and cultivars.

Although cpDNA barcoding offers significant advantages, it is important to consider the limitations of this approach. The relatively slow mutation rate in certain cpDNA regions may limit the resolution for discrimination between closely related cultivars. Therefore, a combination of cpDNA loci with other genetic markers is necessary for optimal cultivar discrimination [34]. There are still challenges in standardizing cpDNA markers and establishing unified protocols for barcoding different plant species in different laboratories, highlighting the need for further improvement of these methodologies.

### 2.3. Nuclear DNA for Cultivars Identification

Among the various DNA regions for plant barcoding, the Internal Transcribed Spacer (ITS) has confirmed its efficiency for many species and was proposed as universal barcode [39,40,41]. The ITS region is a non-coding region located in the ribosomal DNA repeat unit of the nuclear genome [40]. It is flanked by the highly conserved 18S, 5.8S, and 28S ribosomal RNA genes. The ITS region consists of two segments, ITS1 and ITS2, separated by the 5.8S gene. These spacer regions are transcribed but removed during the processing of ribosomal RNA. The ITS region offers several advantages for DNA barcoding applications: (1) high sequence variability, making it suitable for distinguishing closely related species and cultivars [39,40,41,42]; (2) availability of universal primers transferable to a broad range of plant taxa [43]; (3) extensive databases with a large number of ITS sequences [43]; and (4) high success rates in identifying plant species and cultivars using the ITS region [39,44].

Several recent studies have explored the use of ITS barcoding for cultivars discrimination. High discrimination power of ITS sequences was demonstrated in pineapple [44], cassava [41], banana [43], apricot [45], chili pepper [46], and mango [47]. However, studies on pomegranate [48], fig [49], tea plant, and citrus [26] have found the ITS region to be ineffective in distinguishing across cultivars. The limitations of ITS barcoding are as follows: (1) copy number variation: ITS copy number variation can lead to inaccurate phylogenetic inferences; (2) PCR bias: PCR amplification can be biased towards certain ITS variants; (3) hybridization and polyploidy: ITS sequences may not be reliable for identifying hybrids or polyploid species; (4) reference library needs: Kjærandsen [50] highlights the need to build high quality reference libraries; and (5) resource-consuming data analysis: data analysis of sequencing reads is time-consuming to avoid errors and “noises”. These reasons make the unification of protocols in the large-scale experiments difficult. Thus, as compared to the ITS region, cpDNA loci can be a more efficient barcode due to simplicity of amplification, sequencing, and data analysis, as well as greater transferability of primers among different genera.

### 2.4. Multilocus DNA Barcoding Systems for Cultivars Identification

The problem of creating a universal barcode for land plants remains unresolved, since no single locus or combination has been generally accepted. Some researchers have concluded that single-locus barcodes often do not have sufficient resolution to distinguish closely related cultivars or species. In this regard, the use of multilocus DNA barcoding is becoming a popular approach for the accurate identification and discrimination of plant cultivars. For example, the need for a multilocus approach has been highlighted for mango cultivars [51] and closely related clerodendron species [52]. In this study, a combination of the nuclear ITS region with several cpDNA loci (*matK*, *rbcL*, *ycf1*) was evaluated, and the best combination was ITS + matK. This combination was also efficient for grasses (*Agropyron*, *Bromus*, *Elymus*, *Elytrigia*, *Festuca*, *Leymus,* and *Lolium*) [53] and medicinal orchids [54,55]. However, for *Olive* cultivars, matK was the least variable compared to *rbcL*. Variability levels of 77% and 98.7% were observed for matK (fragment of 816 bp) and *rbcL* (the fragment of 411 bp), respectively [56].

Other studies also proposed two-loci DNA barcodes. For example, the effectiveness of a combination of ITS2 and trnH-psbA was confirmed for cultivars discrimination of saffron [57], pomegranate [48], and forage plants [58]. Additionally, the high efficiency of the three-loci barcode (ITS2 + psbA-trnH + trnL-trnF), with an identification rate of 93.6%, was established for nine species of *Syringa*; intraspecific diversity was revealed by each of the loci [59].

Some researchers proposed a combination of traditional genotyping techniques [60,61] and morphological, chemical data [10,62] with the DNA barcoding, leading to more robust variety of identification systems. For example, Wang et al. [63] conducted a comprehensive study of Citri Reticulatae Pericarpium (CRP) using an integrated approach that combined LC-QTOF MS-based metabolomics and DNA barcoding. Their study successfully identified chemical markers that differentiated the Guangchenpi and Chenpi subtypes, demonstrating the effectiveness of combining chemical profiling with molecular methods for cultivars differentiation. Similarly, the development of nomenclatural standards and genotyping methods for potato cultivars [64] highlights the importance of combining traditional taxonomic approaches with modern molecular techniques such as cpDNA analysis.

To summarize, the choice of a strategy for cultivars identification fluctuates between two criteria: (1) labor/time intensity and (2) reliability of identification. Combining different methods is a more labor-intensive approach, limiting its application in the large-scale experiments and making the analysis of a large number of samples difficult. On the other hand, choosing only one or two loci has been associated with the risk of poor reliability of cultivar identification. Despite this, multilocus approaches may become an integral part of modern plant taxonomy and variety identification, ensuring accurate identification and protection of plant genetic resources [54,65,66].

## 3. Selection of Primers

Primer selection is a fundamental step for successful amplification. An important criterion that simplifies the barcoding procedure is the universality of the primers, i.e., their transferability to a large number of plant species and genera. A recent study evaluated the transferability of *rbcL* and *matK* primer sequences in silico using R package “OpenprimeR”. In total, 366 and 489 different primers were found for *rbcL* and *matK*, respectively. These primers were tested in 8463 species. When evaluating the primer-sequence correspondence, the primers with the highest sequence coverage were 96.39% and 93.81% forward and reverse for *rbcL* and 91.56% and 61.62% forward and reverse for *matK*. No universal primer was found for all land plants, but two pairs of *rbcL* primers were able to amplify >99% of the sequences. In contrast to the results obtained for the *matK* region, the 10 pairs optimized for the highest sequence coverage did not cover >85% of the sequences [23].

In silico analysis of existing ITS primers based on highly representative datasets showed that primers universal to this region are suitable for more than 95% of plant species in most groups. A total of 335 samples from 219 angiosperm families, 11 gymnosperm families, 24 fern and lycophyte families, 16 moss families, and 17 fungi families were used to test the performance of these primers [67].

To check the transferability of commonly used primers for cpDNA and for ITS region, we used NCBI Primer Blast (Table 1, Appendix A). The number of accessions for each primer pair varied from 57 (for ITS) to 915 (for trnE-UUC/trnT-GUU) (Figure 2 top). The greatest variability of the product size (dispersion of 3300–3400 bp) was observed for the ITS-targeted primer pairs. The lowest variability of the amplicon length was observed for rbcLa F/rbcLr 590, petB/petD, matK, and 16S (Figure 2 bottom). It seems that there is no correlation between the size of the product and the number of accessions in the NBCI database.

To summarize, large-scale barcoding experiments need the uniformity of protocols, including the application of universal primers, resulting in similar fragment length and highly polymorphic loci. According to the literature data, the most variable cpDNA-loci and primers for intraspecific analysis are proposed: trnE-UUC/trnT-GUU, rpl23/rpl2.l, psbA-trnH, trnL-trnF, trnK, rpoC1, ycf1-a, rpl32-trnL, trnH-psbA, and matK. We suggest that the combination of three of these loci can be a sufficient DNA barcode for cultivar-level identification of agricultural crops (Figure 3). The best combination has to be selected for each crop. The addition of ITS loci can be efficient in some cases. However, it seems that ITS-targeted primers are often not efficient for many species, as they show high level of size variability of the product, which is the constraints for the uniformity of the barcoding procedure. Among the universal chloroplast primers, trnE-UUC/trnT-GUU are transferable for the greatest number of the species and show high polymorphism among closely related genotypes [26].

## 4. Strategies of Amplicon Analysis

### 4.1. Restriction Analysis

The restriction analysis of amplicons leverages the principles of restriction fragment length polymorphism (RFLP) to differentiate between genetic variants within a species, providing a reliable method for cultivar identification and classification. RFLP is a technique that involves the digestion of amplified DNA fragments with specific restriction enzymes that cut the DNA at known sequences. The resulting fragments are then separated by size using gel electrophoresis, allowing for the visualization of distinct patterns that can be used to differentiate between cultivars. For example, Wolff et al. [76] demonstrated that RFLP could effectively distinguish between chrysanthemum cultivars, highlighting the stability of DNA fingerprint patterns. Their study on soybean cultivars revealed that combining endonuclease cleavage with amplification could significantly improve the resolution of genetic markers, facilitating the identification of closely related cultivars. The integration of multiple endonuclease digestions has been shown to enhance the detection of polymorphic DNA [77].

The challenges of RFLP are related to the choice of restriction enzymes, as enzymes that recognize similar sites may yield indistinguishable patterns, complicating the analysis. Additionally, the quality of the PCR amplicons is paramount; impurities can lead to unreliable results. Careful experimental design and validation are necessary to ensure the accuracy of the results [78]. Furthermore, the reproducibility of amplicon sequencing-based detection methods has been a concern.

### 4.2. High Resolution Melting for DNA Barcoding of Cultivars

High-resolution melting (HRM) has emerged as a powerful technique for the identification and authentication of various plant cultivars through DNA barcoding. This method leverages the unique melting profiles of DNA fragments to distinguish between closely related species and cultivars. The melting curve generated provides a unique profile for each DNA sample, allowing for the differentiation of genotypes based on their melting behavior. This technique is sensitive to even minor variations in DNA sequences, making it suitable for detecting single nucleotide polymorphisms (SNPs) and other genetic variations. There are several examples of using HRM for cultivars identification. Muleo et al. [79] demonstrated the effectiveness of HRM analysis in the genotyping of olive germplasm. The study highlighted the method’s ability to detect both homozygous and heterozygous mutations, confirming its high sensitivity and reproducibility for cultivar identification [79]. Jaakola et al. [80] applied HRM analysis for the authentication of bilberry genotypes. In a study by Bosmali et al. [81], HRM analysis was integrated with microsatellite markers and DNA barcoding to distinguish between different lentil varieties and detect admixtures. Madesis et al. [82] reported the application of Bar-HRM analysis for the authentication of bean crops without prior DNA purification. This method proved effective in identifying and quantifying major Greek and Mediterranean bean genotypes [82].

While HRM analysis presents numerous advantages for cultivar identification, challenges remain in optimizing PCR conditions and primer design to maximize melting curve variability. Future research should focus on refining these parameters and exploring the integration of HRM with next-generation sequencing technologies to enhance the resolution and accuracy of cultivar identification.

### 4.3. Sanger Sequencing for Amplicon Analysis

Sanger sequencing, a pioneering technique in the field of molecular biology, has emerged as a valuable tool for cultivars barcoding. Recent studies have underscored the efficacy of Sanger sequencing for amplicon analysis in cultivars barcoding. For instance, the successful application of Sanger sequencing to distinguish various banana cultivars has been demonstrated recently [83]. The researchers employed a combination of universal primers and cultivar-specific primers to amplify and sequence the ITS region, yielding a high degree of accuracy in cultivar identification. Recently, we also efficiently used Sanger sequencing to reveal structural cpDNA loci and an ITS region in four subtropical crops [26]. The results help accurately distinguish the cultivars within each species.

Despite the advances made using Sanger sequencing-based amplicon analysis, contrasting perspectives and challenges persist. The method can be time-consuming and costly, particularly when dealing with large sample sizes. Some researchers have argued that the technique is limited by its relatively low throughput and high cost compared to newer sequencing technologies. However, Sanger sequencing remains a reliable method for initial screenings and for generating high-quality reference sequences. Its high accuracy and reliability make it an indispensable tool for amplicon analysis, particularly in situations where precise cultivar identification is critical [83]. The integration of Sanger sequencing with other molecular techniques can enhance the overall effectiveness of DNA barcoding. For instance, combining Sanger sequencing with high-resolution melting analysis or real-time PCR can improve the accuracy of species identification, particularly in complex samples [84]. Furthermore, the development of multiplex assays, as demonstrated by Richardson et al. [85], allows for the simultaneous identification of multiple species, streamlining the process of cultivar authentication.

### 4.4. Next Generation Sequencing (NGS) for Cultivars Identification

The emergence of NGS has significantly enhanced the efficiency and accuracy of DNA barcoding, enabling the simultaneous sequencing of multiple samples and the generation of vast amounts of data. Meta-barcoding, a technique combining NGS with DNA barcoding, has been successfully applied to complex herbal formulations. Pandit et al. [86] demonstrated the use of *rbcL* gene-based mini-barcodes in high-throughput sequencing to detect plant species in polyherbal products, highlighting NGS’s capacity to analyze multiple species simultaneously. This approach can be adapted for cultivar-level discrimination within plant species, enhancing quality control and authentication processes. Moreover, Shokralla et al. [87] showcased the potential of NGS in enhancing DNA barcode capture from single specimens, allowing for the identification of cryptic species and improving the resolution of taxonomic classifications. NGS facilitates rapid and large-scale sequencing efforts necessary for expanding these reference databases, which are essential for reliable cultivar identification.

Despite the advancements in NGS technologies, several challenges remain in the field of cultivar barcoding. The complexity of plant genomes, particularly in polyploid species, can complicate the interpretation of sequencing data. Additionally, the need for robust reference databases to support accurate species identification is critical. As noted by Antil et al. [88], constructing a comprehensive reference library is essential for the effective application of DNA barcoding. Future research should focus on optimizing NGS protocols to enhance the accuracy and efficiency of cultivar barcoding. The integration of advanced computational tools for data analysis and the development of standardized protocols will be crucial in addressing the current limitations.

#### 4.4.1. Pooled Amplicon Sequencing for Cultivar Identification

Pooled amplicon sequencing has emerged as a powerful tool for cultivar identification, leveraging the advantages of next-generation sequencing (NGS) technologies to provide high-resolution genetic information. This method allows for the simultaneous analysis of multiple samples to distinguish between closely related cultivars. The ability to sequence multiple samples in a single run significantly reduces costs and increases throughput, making it an attractive option for large-scale studies.

Jia et al. [89] emphasized the challenges posed by sequencing errors and cross-contamination in amplicon studies, underscoring the need for robust methodologies to ensure accurate identification of rare taxa. Pooled amplicon sequencing can mitigate these issues by providing a more comprehensive view of genetic diversity. The application of pooled amplicon sequencing in cultivar identification has been demonstrated in various studies. Particularly, Urra et al. [90] applied high-throughput amplicon sequencing to identify grapevine clones, revealing genetic variations among clonal selections of Vitis vinifera cultivars. In rye, pooled sample comprising DNA of 96 individual plants was efficiently used to evaluate for sequence variation in six candidate genes [91]. Our recent study on tea plant showed the efficiency of pooled amplicon sequencing to reveal SNPs and InDels in the amplified genes of different cultivars [92].

Despite its advantages, pooled amplicon sequencing is not without challenges. The choice of target regions and the design of primers can significantly impact the quality and resolution of the data obtained. For instance, Whitford et al. [93] demonstrated that the choice of taxonomic database and variable region of the 16S rRNA gene sequence is critical for achieving accurate species-level identification. Therefore, careful consideration must be given to experimental design to maximize the effectiveness of pooled amplicon sequencing for cultivar identification. Additionally, the potential for sequencing biases, such as those related to GC content and template length, must be addressed. Whitford et al. [94] highlighted the importance of understanding these biases to improve the accuracy of variant discovery using multiplex amplicon sequencing. As the field of genomics continues to evolve, pooled amplicon sequencing is poised to play a pivotal role in cultivar identification and characterization. The integration of advanced sequencing technologies, such as Oxford Nanopore and PacBio, offers the potential for even greater resolution and accuracy in identifying genetic variations among cultivars [95].

#### 4.4.2. Genotyping-by-Sequencing for Cultivars Identification

Genotyping-by-sequencing (GBS) has emerged as a powerful tool for cultivar identification across various plant species. This method leverages NGS technologies to generate SNPs that can be used to assess genetic diversity, establish relationships among cultivars, and facilitate marker-assisted selection in breeding programs.

GBS has been utilized in a variety of crops to enhance the accuracy and efficiency of cultivar identification. For instance, Uitdewilligen et al. [96] used GBS to assess genomic DNA variation in autotetraploid potato cultivars. Their study highlighted the ability to detect numerous sequence variants, which is crucial for cultivar identification and breeding programs. Particularly, Niimi et al. [97] created a set of 10 SNP markers for acid citrus cultivars, enabling the discrimination of 85 different cultivars. Similarly, Park et al. [98] developed a core set of SNP markers for Korean watermelon cultivars. Similarly, Meng et al. [99] utilized GBS to identify commercial cultivars in the Tabebuia alliance, providing a robust basis for patent protection and clarifying the genetic background of these cultivars.

While GBS has proven to be a valuable tool for cultivar identification, challenges remain. The need for high-quality reference genomes and the potential for sequencing errors can complicate data interpretation. However, ongoing advancements in sequencing technologies and bioinformatics tools are expected to address these issues, making GBS an even more powerful resource for plant breeders and geneticists.

#### 4.4.3. RAD-Seq Approach for Cultivar Identification

Recent advancements in molecular techniques, particularly the restriction site-associated DNA sequencing (RAD-seq) approach, have revolutionized the way cultivars are identified. RAD-seq is an NGS-technique that allows for the identification of genetic variation across a wide range of species, including those with limited genomic resources. By focusing on specific regions of the genome, RAD-seq generates a large number of single nucleotide polymorphisms (SNPs) that can be used for cultivar identification. This method is particularly advantageous for non-model organisms, where reference genomes may not be available.

The application of RAD-seq has been demonstrated in various plant species, including lavender [100], ryegrass [101], and rhododendron [102]. For instance, a study on Italian ryegrass (Lolium multiflorum) utilized RAD-seq to distinguish between 11 varieties, revealing significant genetic differences [101]. Similarly, in tobacco, RAD-seq was employed to develop insertion–deletion (InDel) markers, which are valuable for genetic studies and marker-assisted selection breeding [103]. The results indicated that RAD-seq could effectively capture the genetic intraspecific diversity, facilitating accurate cultivars identification. Comparative studies have highlighted the advantages of RAD-seq over other genomic approaches [104]. The combination of RAD-seq with machine learning algorithms has shown promise in accurately classifying cultivars based on spectral data, as demonstrated in studies involving cotton [105] and olive [106] cultivars. These approaches leverage the strengths of both molecular and computational techniques to improve identification accuracy.

A critical analysis of RAD-sec perspectives reveals that the choice of restriction enzyme, sequencing depth, and data analysis pipeline can significantly impact the accuracy of RAD-seq-based cultivar identification. As we move forward, future research directions should focus on optimizing RAD-seq protocols for specific crops, exploring the integration of RAD-seq with other omics technologies and developing more sophisticated data analysis tools to fully harness the power of this approach.

#### 4.4.4. Oxford Nanopore Technologies for Cultivars Identification

Oxford Nanopore Technologies (ONT) has emerged as a versatile platform for various genomic applications, including cultivar barcoding, owing to its capacity for long-read sequencing, and portability. One of the foundational advantages of ONT for cultivar barcoding is its capacity to produce long reads that encompass entire barcode regions, such as the ITS and chloroplast genomes, which are critical for species and cultivar identification.

The ability to generate comprehensive genomic datasets using ONT has been demonstrated in chloroplast genome sequencing, which is often used as a plant barcode. Wahyuni et al. [107] successfully assembled a draft chloroplast genome of *Dryobalanops aromatica* using ONT data. Their work highlights the potential of ONT to produce genome-scale data that can be used for detailed phylogenetic and cultivar differentiation analyses. Similarly, Aury et al. [108] reported on the assembly of the hexaploid wheat genome using long reads from ONT, achieving high-resolution assemblies suitable for research and breeding. These studies underscore the capacity of ONT to generate high-quality, long-read data that are instrumental in resolving complex plant genomes and distinguishing cultivars at the genomic level.

The body of research underscores the significant potential of Oxford Nanopore sequencing in plant cultivar barcoding. Its ability to generate long reads covering entire barcode regions, coupled with advanced error correction and demultiplexing tools, makes it a powerful platform for rapid, cost-effective, and high-throughput cultivar identification. As bioinformatics tools continue to evolve, the accuracy and scalability of ONT-based barcoding are expected to improve further, solidifying its role in plant genetics, breeding, and conservation efforts. The versatility demonstrated across various studies indicates that ONT is well-positioned to become a standard tool in the molecular identification and phylogenetic analysis of cultivars.

#### 4.4.5. Application of Whole-Genome Resequencing as Super-Barcode

In recent years, whole genome resequencing has taken center stage, allowing for the comprehensive analysis of a plant’s genetic makeup. This has become possible due to development of NGS technologies. However, the whole genome resequencing of the nuclear genome is prohibitively expensive for widespread adoption, particularly in resource-limited laboratories. A critical analysis of these perspectives reveals that the key to successful implementation lies in balancing technological advancement with economic viability. As the cost of whole genome resequencing continues to decrease, its application in cultivar identification is likely to become more feasible [109]. Furthermore, the development of more efficient bioinformatic tools and pipelines will be essential for analyzing the vast amounts of data generated by whole genome resequencing [110]. As this technology continues to evolve, it is likely to play an increasingly vital role in agriculture and horticulture, enabling the precise identification and conservation of valuable crop cultivars.

The whole resequencing of the chloroplast and mitochondrial genomes is often used as an efficient approach for DNA-barcoding on intraspecific diversity. Recent barcoding studies have placed high emphasis on the use of whole-chloroplast genome sequences as a “super-barcode”, which has become increasingly feasible due to advancements in sequencing technologies [111]. For instance, Zhang et al. [112] conducted a comparative analysis of the chloroplast genome of *Lonicera japonica* cultivars. Whole-genome sequencing of cpDNA has been used to identify tea (*Camellia sinensis*) cultivars, providing valuable phylogenomic information [113]. In addition, complete chloroplast genome sequences have been used to identify the medicinal plant *Scutellaria baicalensis* [35]. For the authentic rice genotypes, the whole chloroplast genome trnL-F region was the most reliable barcode, although it required extensive sequencing and informatic analyses [114]. While whole chloroplast genome sequencing can already deliver a reliable meta-barcode for accurate plant identification, it is not yet resource-effective and does not yet offer the speed of analysis provided by single-locus barcodes to unspecialized laboratory facilities [33].

## 5. Bioinformatic Tools for Barcoding Data Analysis

The analysis of DNA barcoding data requires robust bioinformatic tools that can handle large datasets generated from sequencing technologies. These tools assist in various stages of the barcoding process, including sequence alignment, phylogenetic analysis, and data visualization. Bioinformatic tools such as MEGA (Molecular Evolutionary Genetics Analysis) and Clustal Omega are commonly used for sequence alignment, allowing researchers to compare genetic sequences across different species. However, in the context of plant barcoding, large datasets can provide valuable insights into evolutionary processes and species diversity, especially in complex groups with polyploidy and hybridization [115]. This limitation underscores the need for tools that can efficiently calculate sequence divergences and perform analyses without the computational restrictions typical of multiple alignment algorithms [116]. In the realm of metabarcoding, a variety of bioinformatic tools have been evaluated to address the unique challenges posed by high-throughput sequencing data. The recent review underscores the importance of pipelines that can effectively process reference sequence data, with many tools designed to facilitate the transition from raw data to ecological insights [117]. The metabaR package further contributes to this field by providing an R-based solution aimed at evaluating and improving DNA metabarcoding datasets, although it highlights the ongoing need for tools that seamlessly integrate with ecological analysis workflows [118]. In addition, several bioinformatic tools have been developed to address the challenges associated with large barcoding datasets:

SeqTrace is an open-source software which can identify, align, and compute consensus sequences from matching forward and reverse traces, filter low-quality base calls, and end-trim finished sequences [119]. The software features a graphical interface that includes a full-featured chromatogram viewer and sequence editor. However, SeqTrace needs manual assistance in many sequence analysis tasks and can often generate mismatches and gaps in the final consensus sequences, reducing the reliability of the results.

PIPEBAR was later developed to overcome these problems. This pipeline works with Sanger sequencing chromatograms and allows us to run barcode analysis of hundreds of sequences in a fast, accurate, and concise command line [120]. PIPEBAR is faster than other software and can be used to facilitate the submission of barcode sequences to databases such as BOLD and NCBI.

AmpliPiper, a versatile analysis pipeline, is designed explicitly for multilocus amplicon sequencing data generated through third-generation sequencing technologies such as Oxford Nanopore. This tool automates bioinformatics workflows, enabling researchers to manage large datasets effectively and extract meaningful taxonomic and biodiversity insights [95]. Its user-friendly interface and automation capabilities address the challenges posed by high-throughput sequencing, making it suitable for large-scale biodiversity assessments.

DNA Subway is a platform that integrates research-grade bioinformatics software into streamlined workflows, facilitating sequence analysis and data interpretation [121]. This approach simplifies the process for researchers by bundling essential tools into user-friendly pipelines, thereby enhancing accessibility and efficiency. Similarly, the Cogent NGS Analysis Pipeline exemplifies a comprehensive framework capable of handling diverse sequencing data types, including gene, single-cell DNA, and other NGS datasets, with functionalities tailored for gene identification and functional annotation [Cogent™ NGS Analysis Pipeline].

Multiplexing and demultiplexing samples are vital for high-throughput cultivar barcoding. Sample demultiplexing is supported by tools such as those developed by 10× Genomics, which utilize unique DNA barcodes for sample identification. These tools enable efficient separation of pooled samples, ensuring accurate downstream analysis [https://www.10xgenomics.com/analysis-guides/bioinformatics-tools-for-sample-demultiplexing (accessed on 13 July 2025)]. Han et al. [122] developed HycDemux, a hybrid unsupervised approach that improves demultiplexing accuracy by integrating nanopore signals and DNA sequences. This approach outperforms existing tools, especially in complex multi-sample datasets, making it suitable for large-scale cultivar barcoding projects. Similarly, Papetti et al. [123] presented UNPLEX, a framework that operates directly on raw nanopore signals for barcode demultiplexing, further enhancing the efficiency and accuracy of sample identification. Additionally, the development of versatile frameworks like mBARq demonstrates efforts to create user-friendly analysis environments capable of handling barcoded sequencing data, although the current tools often lack the flexibility required for diverse experimental designs [124].

Error correction and consensus sequence generation are critical for accurate cultivar barcoding, given the higher error rates associated with ONT sequencing compared to short-read platforms. Sahlin et al. [125] introduced NGSpeciesID, a software tool designed to produce highly accurate consensus sequences from long-read amplicon data, including ONT reads. This tool minimizes preprocessing and enhances scalability, enabling rapid processing of large sample sets while maintaining high accuracy. Such tools are essential for reliable cultivar identification, as they improve the usability of ONT data for barcode-based discrimination. Rafeie et al. [126] introduced Medlib, a high-performance alignment library for ONT tailored for noisy long-read data, facilitating accurate sequence alignment necessary for barcode analysis. Such tools are vital for ensuring the reliability of ONT-based cultivar barcoding workflows.

For forensic applications, specialized bioinformatic pipelines, such as NGSpeciesID, have been developed to process Nanopore sequencing data for species identification, emphasizing the importance of tailored solutions for specific use cases [127]. The inclusion of profile-hidden Markov models (HMMs) in sequence analysis further enhances the accuracy of barcode data interpretation by filtering pseudogene sequences that could otherwise lead to misleading results [128].

VAREANT is a bioinformatics application designed to streamline the preparation of genomic variant data. It comprises three modules: pre-processing, variant annotation, and AI/ML data preparation. This tool simplifies the workflow for curating targeted variant datasets, making it particularly useful for researchers dealing with large genomic datasets [129]. By enabling efficient data preparation, VAREANT supports the identification of novel biomarkers and the stratification of patients based on disease risk factors.

Taxonize-gb is a command-line software tool designed to filter GenBank non-redundant databases based on taxonomy. This tool allows researchers to create taxa-specific reference databases tailored to their research questions, significantly reducing search times and improving the efficiency of data analysis [130]. By enabling the extraction of relevant sequences, Taxonize-gb enhances the accuracy of species identification in metagenomic studies.

QuasiFlow is another bioinformatic tool that focuses on analyzing genetic variability from NGS data. It is particularly useful for studying quasispecies, which are populations of closely related viral genomes. QuasiFlow can extract reliable mutations and recombinations, even at low frequencies, making it a valuable resource for researchers investigating genetic diversity within species [131]. This capability is essential for understanding evolutionary dynamics and species adaptation.

The superSTR tool offers an alignment-free method for detecting repeat expansions in DNA and RNA sequencing data. This ultrafast tool is capable of efficiently processing large datasets, making it suitable for applications in population genetics [132]. Its ability to identify repeat expansion motifs can complement traditional DNA barcoding approaches by providing additional genetic information relevant to species identification.

Despite the advancements in bioinformatic tools for DNA barcoding analysis, several challenges remain. The complexity of plant genomes, which often contain large amounts of repetitive sequences, can complicate data analysis and interpretation. Additionally, the need for standardized protocols and databases is critical for ensuring the reliability and reproducibility of barcoding results. Furthermore, as sequencing technologies continue to evolve, bioinformatic tools must adapt to handle the increasing complexity and volume of data generated. Future developments in bioinformatics, particularly in machine learning and artificial intelligence, hold great promise for enhancing the analysis of DNA barcoding data. These technologies can improve the accuracy of species identification and facilitate the discovery of new plant species by analyzing vast datasets more efficiently [133].

## 6. Barcoding Databases and Libraries

The accuracy of DNA barcoding heavily depends on well-curated reference libraries [134]. Their study highlights the necessity of incorporating expert validation and comprehensive metadata to ensure the reliability of large datasets, which in turn influences the performance of bioinformatics tools used for species identification. In 2003, the first DNA barcode research center was established in Canada [135]. Now, there are research services empowered by more than USD 7 million in sequencing and liquid handling instrumentation and more than 20 expertly skilled staff (CCDB. http://dnabarcoding.ca/, accessed on 13 July 2025). The mission of the CCDB is to integrate DNA barcoding technology within their research programs and day-to-day operations. The development of cpDNA genetic databases demonstrates the growing efforts towards standardized data collection and sharing. Nowadays, there are several, widely used DNA barcode databases. The Barcode of Life Data System (BOLD, http://boldsystems.org/, accessed on 13 July 2025) is a central resource for biodiversity science aimed at the acquisition, storage, validation, analysis, and publication of DNA barcodes [136]. However, this platform is mainly used for the identification of species and genera. On the other hand, the platform for intraspecific diversity of local crop collections worldwide is of crucial importance for the sustainability of agricultural systems.

There are several examples of national barcoding libraries. Particularly, in China, a DNA barcode identification system was developed for the identification of herbal plants; it was established based on a two-locus combination of ITS2+psbA–trnH barcodes (http://www.tcmbarcode.cn/china/, accessed on 13 July 2025). In addition, a comprehensive DNA barcoding library for woody plants was developed for the conservation and management of tropical and subtropical forests [137]. The dataset includes a standard barcode library comprising the four most widely used barcodes (*rbcL*, *matK*, ITS, and ITS2) for 2520 species from 4654 samples. In the UK, a national DNA barcoding reference library was created based on Sanger sequencing of *rbcL*, *matK*, and ITS2, and it contains 1482 plant species. Species-level discrimination was highest with ITS2; however, the ability to successfully retrieve a sequence from herbarium samples was lowest for these loci [2]. In Peru, a DNA barcode reference library of the Lomas region was established based on *rbcL*, *matK*, and ITS2 loci. This database provides 1207 plant specimens of 16 Lomas locations in Peru [138]. In Greece, the ‘Greek Vitis Database’ database gathers information about Greek cultivars of *Vitis vinifera* and highlights a multifaceted approach to cultivar characterization [18].

## 7. Conclusions

To summarize, the selection of appropriate loci that can discriminate between closely related varieties remains a critical aspect of developing effective identification systems for the cultivars. Based on the results of our review, we propose to use a combination of 3–4 chloroplast loci (trnE-UUC/trnT-GUU, rpl23/rpl2.l, psbA-trnH, trnL-trnF, trnK, rpoC1, ycf1-a, rpl32-trnL, trnH-psbA and *matK*); this combination has to be selected for each particular crop. The use of ITS region to complete this combination can be useful in some cases. This approach can be efficient for large-scale barcoding efforts. The Sanger sequencing can be a reliable method of amplicon analysis in such experiments. On the other hand, NGS technologies and whole resequencing of chloroplast and mitochondrial genomes can be reliable approaches to develop super barcodes of elite varieties. The development of comprehensive barcode reference libraries for the local germplasm collections is expected to increase the throughput and resolution of the barcoding methods [139]. In addition, automatization and standardization of barcoding protocols are considered as methods to facilitate the large-scale identification and certification of the intraspecific diversity of plants. While the application of barcoding to plant varieties is still evolving, advances in medical barcoding and macroalgal research demonstrate the versatility and expanding scope of DNA-based sample identification methods [140].

## Figures and Tables

**Figure 1 ijms-26-06808-f001:**
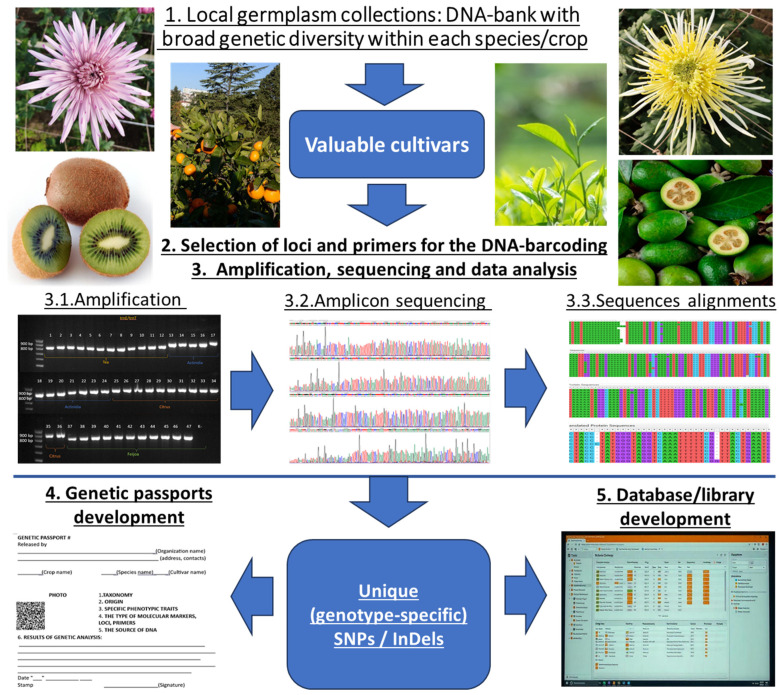
Key stages of DNA barcoding of cultivars in the local germplasm collections: selection of loci and primers, amplificons sequencing, alignments, data analysis to reveal genotype-specific (unique) SNPs and InDels, check the specificity on broad range of intraspecific diversity, develop genetic passport of the cultivar, create DNA-library/database.

**Figure 2 ijms-26-06808-f002:**
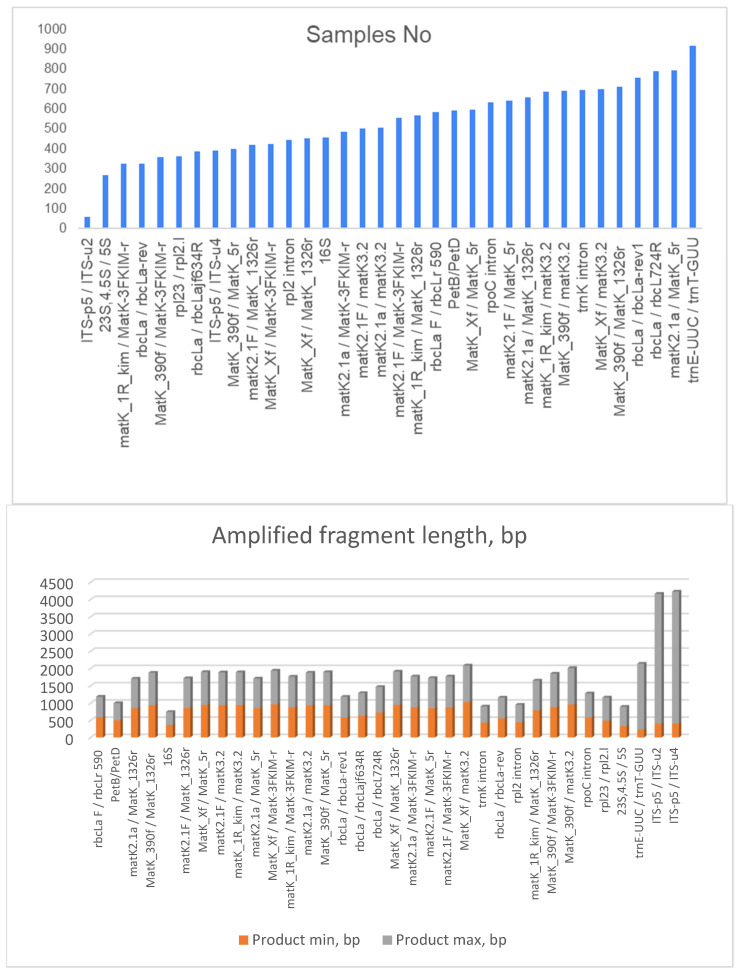
NCBI Primer blast results on universal chloroplast primers and ITS primers. **Top barplot**—the number of accessions in NCBI database, **Bottom barplot**—the length of amplified product (min and max) for each primer pair.

**Figure 3 ijms-26-06808-f003:**
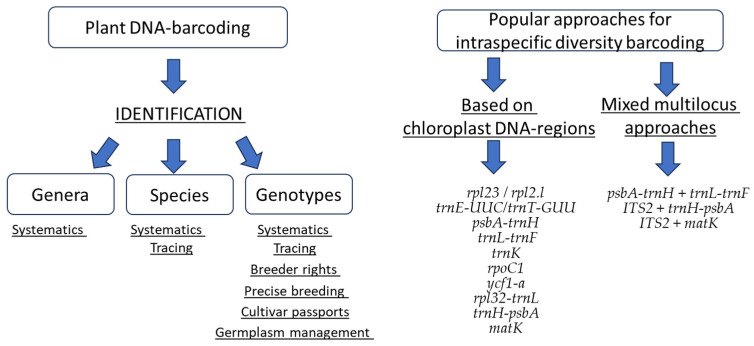
Application of plant DNA barcoding and the most frequently used DNA barcode loci for genotype-level identification of valuable plant crops. The left scheme presents the levels of plant identification using DNA barcoding and application of the results. The right scheme represents the most efficiently used loci and combinations for genotype intraspecific genotype-level identification.

**Table 1 ijms-26-06808-t001:** Universal primers for DNA-barcoding of plant species.

Primer Code	Forward/Reverse	Sequence 5′-3′	Reference
rbcLa-F	F	ATGTCACCACAAACAGAGACTAAAGC	[20]
rbcLr590	R	AGTCCACCGCGTAGACATTCAT	[68]
rbcLa-rev	R	GTAAAATCAAGTCCACCRCG	[69]
rbcLajf634R	R	GAAACGGTCTCTCCAACGCAT	[70]
rbcL724R	R	TCGCATGTACCTGCAGTAGC	[71]
matK2.1a	F	ATCCATCTGGAAATCTTAGTTC	[72]
matK2.1F	F	CCTATCCATCTGGAAATCTTAG	[72]
matK_1R_kim	F	ACCCAGTCCATCTGGAAATCTTGGTCC	[73]
MatK_390f	F	CGATCTATTCATTCAATATTTC	[74]
MatK_Xf	F	TAATTTACGATCAATTCATTC	[72]
MatK-3FKIM-r	R	CGTACAGTACTTTTGTGTTTACGAG	[73]
MatK_1326r	R	TCTAGCACACGAAAGTCGAAGT	[74]
MatK_5r	R	GTTCTAGCACAAGAAAGTCG	[72]
matK3.2	R	CTTCCTCTGTAAAGAATTC	[72]
23S,4.5S/5S	F	TCTCCTCCGACTTCCCTAG	[22]
23S,4.5S/5S	R	ACCATGAACGAGGAAAGGC	[22]
16S	F	ATTGCGTCGTTGTGCCTGG	[22]
16S	R	GATACGTTGTTAGGTGCTCC	[22]
petB/petD	F	TAGGGGGAATTACACTTAC	[22]
petB/petD	R	CATTAACATGAATACGGCAG	[22]
rpl23/rpl2.l	F	GAAGAAGCTTGTACAGTTTGG	[22]
rpl23/rpl2.l	R	TTTACTTACGGCGACGAAG	[22]
rpl2 intron	F	ATTGAGTTCAGTAGTTCCTC	[22]
rpl2 intron	R	CCAAACTGTACAAGCTTCTTC	[22]
rpoC1 intron	F	GAGTAACATGAAGCTCAG	[22]
rpoC1 intron	R	GTTTCCTTTCATCCGGCT	[22]
trnK intron	F	GTCTACATCATCGGTAGAG	[22]
trnK intron	R	CAACCCAATCGCTCTTTTG	[22]
trnE-UUC/trnT-GUU	F	TCCTGAACCACTAGACGATG	[75]
trnE-UUC/trnT-GUU	R	ATGGCGTTACTCTACCACTG	[75]
ITS-p5	F	CCTTATCAYTTAGAGGAAGGAG	[67]
ITS-p3	F	YGACTCTCGGCAACGGATA	[67]
ITS-u4	R	RGTTTCTTTTCCTCCGCTTA	[67]
ITS-u2	R	GCGTTCAAAGAYTCGATGRTTC	[67]

## Data Availability

Not applicable.

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
