# Peer review of "DNA-Barcoding for Cultivar Identification and Intraspecific Diversity Analysis of Agricultural Crops"

_ijms, 2025, doi:10.3390/ijms26146808_

Round 1
Reviewer 1 Report
Comments and Suggestions for Authors
Dear authors
Thank you for the opportunity to review this manuscript. The paper addresses an important and timely topic, providing an overview of DNA barcoding strategies for cultivar-level identification. However, in its current form, the review is primarily descriptive and would benefit from more depth, systematic comparison, and practical recommendations. I encourage the authors to address the following points to improve the manuscript:
-
Figure presentation and numbering:
Please carefully check the figure numbering throughout the manuscript. Ensure that the workflow diagram (currently Figure 1) and the conceptual framework (currently Figure 2) are correctly labelled and consistently cited in the text. Expand the figure captions to clearly explain what each figure shows, including relevant loci, sample species, and the role of each step in the workflow. -
Add comparative tables:
The manuscript currently lacks any tables summarizing the characteristics and performance of different barcoding loci. It would greatly enhance the usefulness of the review to include at least one or two tables comparing loci in terms of marker type (chloroplast, nuclear, multilocus), resolution for intraspecific diversity, typical target crops, and key references. Such tables will allow readers to quickly assess which loci are suitable for specific applications. -
Clarify literature review method:
Even for a narrative review, it is important to explain how the literature was selected. Please provide details on the databases searched, keywords used, time frame covered, and any inclusion or exclusion criteria applied. This will help readers evaluate the comprehensiveness of the review and avoid potential bias. -
Deepen critical analysis and practical framework:
Currently, the discussion largely summarizes existing studies without proposing new perspectives or actionable guidance. I encourage the authors to identify specific gaps in current barcoding practice, suggest a recommended panel of loci for key crop groups, and outline practical steps for building genetic passports. A clear framework would add significant value and distinguish this review from previous summaries. -
Improve clarity and language:
The manuscript includes some redundant phrases and inconsistent terminology. Please revise the text thoroughly to ensure clarity, logical flow, and a consistent academic tone. Professional language editing would be helpful to polish the writing and enhance readability. -
Address minor content and formatting issues:
Consider shortening the abstract to focus on the main findings and key contributions. Refine the keywords to include more precise terms directly related to the main topic (e.g., “genetic passport”, “cultivar authentication”). Ensure that all references are up-to-date and consistently formatted according to the journal’s style. Lastly, improve the resolution of any images (e.g., gel electrophoresis) and include appropriate scale bars or labels if needed.
I believe that addressing these points will substantially improve the quality and impact of the manuscript. I recommend a major revision.
Comments on the Quality of English LanguageThe manuscript is generally understandable but contains redundant phrases, inconsistent terminology, and occasional awkward sentence structures. I recommend a thorough language revision to improve clarity and ensure a more precise academic tone. Professional English editing would be beneficial before resubmission.
Author Response
Comment 1: [Figure presentation and numbering: Please carefully check the figure numbering throughout the manuscript. Ensure that the workflow diagram (currently Figure 1) and the conceptual framework (currently Figure 2) are correctly labelled and consistently cited in the text. Expand the figure captions to clearly explain what each figure shows, including relevant loci, sample species, and the role of each step in the workflow.]
Response 1: [We have added new figures and tried to expand the figure legends]
Comment 2 : [Add comparative tables: The manuscript currently lacks any tables summarizing the characteristics and performance of different barcoding loci. It would greatly enhance the usefulness of the review to include at least one or two tables comparing loci in terms of marker type (chloroplast, nuclear, multilocus), resolution for intraspecific diversity, typical target crops, and key references. Such tables will allow readers to quickly assess which loci are suitable for specific applications.]
Response 2: [We have added new table 1 in the text, also Supplementary table is attached to the article]
Comment 3: [Clarify literature review method: Even for a narrative review, it is important to explain how the literature was selected. Please provide details on the databases searched, keywords used, time frame covered, and any inclusion or exclusion criteria applied. This will help readers evaluate the comprehensiveness of the review and avoid potential bias.]
Response 3: [We have added Materials and methods previously. It was mentioned in MM section as follows: “The collection of the relevant information was performed using Google search, ResearchGate search. The key words for the literature collection were: “cultivars barcoding”, “chloroplast DNA cultivars barcoding”, “mitochondrial DNA cultivars barcoding”, “plant barcoding intraspecific diversity”, “cultivars certification”, “ plant barcoding for genetic passports”, “plant barcoding database”, “cultivars barcoding database”, “universal primers for cultivars barcoding”. NCBI search was used to find universal primers for cultivars barcoding. The specificity of these primers was checked using NCBI Primer Blast. Totally 4 combinations of rbcL primers and 20 combinations of matK primers were included in the in silico specificity check. In addition, the specificity of 8 pails of the other universal chloroplast primers were analyzed, including intergenic regions (23S,4.5S/ 5S, 16S, petB / D, rpl23 / rpl2.l, rpl2 intron, rpoC1 intron, trnK intron, trnE-UUC/trnT-GUU) and three pairs of primers for nuclear barcode (ITS-p5/ITS-u4, ITS-p5/ITS-u2, ITS-p3/ITS-u4) [17]. The lists of the species were generated for each primer pairs and presented in supplementary file. The focus was on those primers which reported to reveal intraspecific polymorphism in many crops. In addition, AI tool PaperDigest https://www.paperdigest.org/ was used to complete the results with the 5-years old studies. The short reviews were performed using the filters “5 years old” and the following search topics: “chloroplast DNA cultivars barcoding”, “mitochondrial DNA cultivars barcoding”, “multilocus approaches for cultivars barcoding”, “Locus selection for cultivars barcoding”. The obtained results were filtered to remove too general sentences and were combined with those obtained by Google search and ResearchGate search.”
However, the handling editor said, that review papers of IJMS are not included this section, thus we deleted this chapter.]
Comment 4: [Deepen critical analysis and practical framework: Currently, the discussion largely summarizes existing studies without proposing new perspectives or actionable guidance. I encourage the authors to identify specific gaps in current barcoding practice, suggest a recommended panel of loci for key crop groups, and outline practical steps for building genetic passports. A clear framework would add significant value and distinguish this review from previous summaries.]
Response 4: [The panel of loci is recommended in fig 3, the practical steps are added in fig 1. Also , each chapter contain future research direction which are relevant to achieve efficient barcoding system.]
Comment 5: [Improve clarity and language: The manuscript includes some redundant phrases and inconsistent terminology. Please revise the text thoroughly to ensure clarity, logical flow, and a consistent academic tone. Professional language editing would be helpful to polish the writing and enhance readability.]
Response 5: [We have improved the language style and readability of the article. Please check.]
Comment 6: [Address minor content and formatting issues: Consider shortening the abstract to focus on the main findings and key contributions. Refine the keywords to include more precise terms directly related to the main topic (e.g., “genetic passport”, “cultivar authentication”). Ensure that all references are up-to-date and consistently formatted according to the journal’s style. Lastly, improve the resolution of any images (e.g., gel electrophoresis) and include appropriate scale bars or labels if needed.].
Response 6: [The keywords are revised accordingly. The abstract is revised, to make it more focused. The reference list is revised and formatted. The figures are improved.]

Reviewer 2 Report
Comments and Suggestions for Authors
The manuscript presents a timely and relevant review on the application of DNA barcoding for identifying intraspecific diversity in agricultural crops, focusing on its role in developing genetic passports for cultivars. The topic is highly relevant to current challenges in agriculture, including cultivar authentication, intellectual property protection, and ensuring food quality and traceability.
While a good starting point, the manuscript's quality is very low regarding the use of English and meaning is lost across the text, along with the depth of presenting.
While the review covers a broad range of topics, the narrative does not flow logically from one point to the next and some sections lack clarity and transitions between paragraphs and subtopics fail.
Also, does the review aim at specialists in DNA barcoding or a broader audience interested in agricultural crops? It is not clear enough.
The methodological discussion is lacking depth. The review could more explicitly differentiate the challenges and specific marker requirements for intraspecific identification compared to interspecific (species-level) barcoding. Many standard barcoding markers (rbcL, matK) are often insufficient for intraspecific variation, and this limitation should be more strongly emphasized.
While SSRs and SNPs are mentioned, a deeper dive into the power of NGS technologies like genotyping-by-sequencing, RAD-seq, and whole-genome resequencing for discovering SNPs in crops would be highly beneficial.
DNA barcoding along with NGS data generates massive datasets, and bioinformatics challenges like data processing, variant calling, population structure analysis, is crucial, as these are often a major obstacle.
Ensure all technical terms and acronyms are either clearly defined upon first use.
A more critical analysis of why or how certain approaches worked well for specific crops and their limitations would enhance the review's depth. Also, beyond the molecular aspects, discuss the practical challenges of implementing DNA barcoding systems for cultivar identification on a large scale cost, time, regulation , standardization across different labs.
While "protection of breeders' copyrights" is mentioned, expanding on the economic benefits for farmers, consumers, and the seed industry would strengthen the justification for these technologies.
The "future prospects" section could be expanded to include more cutting-edge or speculative technologies, such as Oxford Nanopore sequencers for on-site identification, or the integration of DNA barcoding with machine learning for more in depth crop characterization.
While focusing on the last five years, ensure that the selected references adequately represent the state-of-the-art across different crop types and DNA barcoding advancements. Include examples from a wider range of agricultural crops to demonstrate broader applicability.
By addressing these points, the manuscript can evolve into a more robust, critical, and comprehensive review that not only summarizes the current state but also provides deeper insights into the challenges and future directions of DNA barcoding for intraspecific diversity in agricultural crops.
Comments on the Quality of English LanguageMeaning is lost due to the wrong use of English language. Repetitive phrasing. Major English revisions are required. A thorough proofreading for grammatical errors, typos, or awkward phrasing is needed to improve readability.
Author Response
Comment 1: [While a good starting point, the manuscript's quality is very low regarding the use of English and meaning is lost across the text, along with the depth of presenting.]
Response 1: [We revised the manuscript in accordance with this suggestion.]
Comment 2: [While the review covers a broad range of topics, the narrative does not flow logically from one point to the next and some sections lack clarity and transitions between paragraphs and subtopics fail.]
Response 2: [We revised the manuscript and added the transitions between paragraphs.]
Comment 3: [Also, does the review aim at specialists in DNA barcoding or a broader audience interested in agricultural crops? It is not clear enough.]
Response 3: [The review aimed the germplasm holders, breeding researchers, and other specialists in agricultural crops, who are interested on germplasm characterization and management.]
Comment 4: [The methodological discussion is lacking depth. The review could more explicitly differentiate the challenges and specific marker requirements for intraspecific identification compared to interspecific (species-level) barcoding. Many standard barcoding markers (rbcL, matK) are often insufficient for intraspecific variation, and this limitation should be more strongly emphasized.]
Response 4: [we have tried to revise the chapter accordingly. We added the summarize table and figure of the efficiency of different loci, we have proposed the most suitable combinations for the barcoding of intraspecific diversity]
Comment 5: [While SSRs and SNPs are mentioned, a deeper dive into the power of NGS technologies like genotyping-by-sequencing, RAD-seq, and whole-genome resequencing for discovering SNPs in crops would be highly beneficial.]
Response 5: [We have added the separate chapter of the methods of amplicons analysis.}
Comment 6: [DNA barcoding along with NGS data generates massive datasets, and bioinformatics challenges like data processing, variant calling, population structure analysis, is crucial, as these are often a major obstacle.]
Response 6: [We have added a chapter on bioinformatic analysis]
Comment 7: [Ensure all technical terms and acronyms are either clearly defined upon first use.]
Response 7: [revised]
Comment 8: [ more critical analysis of why or how certain approaches worked well for specific crops and their limitations would enhance the review's depth. Also, beyond the molecular aspects, discuss the practical challenges of implementing DNA barcoding systems for cultivar identification on a large scale cost, time, regulation , standardization across different labs.]
Response 8: [We have added the discussion of these points in the chapter 3 and 4]
Comment 9: [While "protection of breeders' copyrights" is mentioned, expanding on the economic benefits for farmers, consumers, and the seed industry would strengthen the justification for these technologies.]
Response 9: [I’m not sure that I clearly understand, what exactly should we add in the article?]
Comment 10: [The "future prospects" section could be expanded to include more cutting-edge or speculative technologies, such as Oxford Nanopore sequencers for on-site identification, or the integration of DNA barcoding with machine learning for more in depth crop characterization.]
Response 10: [As soon as article composes several blocks now, we added future prospects in the end of each block.]
Comment 11: [While focusing on the last five years, ensure that the selected references adequately represent the state-of-the-art across different crop types and DNA barcoding advancements. Include examples from a wider range of agricultural crops to demonstrate broader applicability.]
Response 11: [The manuscript is significantly expanded and completed with many new references.]
Comment 12: [Meaning is lost due to the wrong use of English language. Repetitive phrasing. Major English revisions are required. A thorough proofreading for grammatical errors, typos, or awkward phrasing is needed to improve readability.]
Response 12: [ the English of whole manuscript is carefully revised]
Round 2
Reviewer 1 Report
Comments and Suggestions for Authors
The revised manuscript has addressed the major concerns raised in the previous round. The authors have made substantial improvements, and the overall structure, scientific content, and clarity are now acceptable for publication. I recommend acceptance of the manuscript in its current form.
Reviewer 2 Report
Comments and Suggestions for Authors
I would like to thank the authors for respecting and implementing my comments.
I believe that the manuscript in its current form represents a more comprehensive review of DNA barcoding for agricultural crops as it covers various aspects, including different DNA regions, sequencing technologies, bioinformatic tools, and databases, emphasizing the importance of DNA barcoding for developing genetic passports, protecting breeders' rights, and preventing counterfeit products.
Overall, the manuscript now is well-structured and more informative.